# Are Parent-Held Child Health Records a Valuable Health Intervention? A Systematic Review and Meta-Analysis

**DOI:** 10.3390/ijerph16020220

**Published:** 2019-01-14

**Authors:** Muhammad Chutiyami, Shirley Wyver, Janaki Amin

**Affiliations:** 1Faculty of Human Sciences, Macquarie University, Sydney, NSW 2109, Australia; shirley.wyver@mq.edu.au; 2Faculty of Medicine and Health Sciences, Macquarie University, Sydney, NSW 2109, Australia; janaki.amin@mq.edu.au

**Keywords:** child health, maternal and child health, parent-held record, effectiveness, parent views, child outcomes

## Abstract

Parent-held child health record (PHCHR), a public health intervention for promoting access to preventive health services, have been in use in many developed and developing countries. This review aimed to evaluate the use of the records toward promoting child health/development. We searched PubMed, PsycINFO, CINAHL, Cochrane Library and Google Scholar to identify relevant articles, of which 32 studies met the inclusion criteria. Due to considerable heterogeneity, findings were narratively synthesised. Outcomes with sufficient data were meta-analysed using a random-effects model. Odds Ratio (OR) was used to compute the pooled effect sizes at 95% confidence interval (CI). The pooled effect of the PHCHR on the utilisation of child/maternal healthcare was not statistically significant (OR = 1.31, 95% CI 0.92–1.88). However, parents who use the record in low- and middle-income countries (LMIC) were approximately twice as likely to adhere to child vaccinations (OR = 1.93, 95% CI 1.01–3.70), utilise antenatal care (OR = 1.60, 95% CI 1.23–2.08), and better breastfeeding practice (OR = 2.82, 95% CI 1.02–7.82). Many parents (average-72%) perceived the PHCHR as useful/important and majority (average-84%) took it to child clinics. Health visitors and nurses/midwives were more likely to use the record than hospital doctors. It is concluded that parents generally valued the PHCHR, but its effect on child health-related outcomes have only been demonstrated in LMIC.

## 1. Introduction

Parent-held child health record (PHCHR), a means of promoting access to preventive health services, involves giving families an ongoing record used in monitoring children health and development [1]. Many developed countries have a long history of use of the PHCHR. For instance, in South Australia, these records are in use from 1981 [1], while in Japan, it has been in use since 1947, which was associated with significant decrease in infant mortality rate [2]. A growing number of developing countries, such as Kenya [3], Indonesia [4], Mongolia [5] and Bosnia [6], have implemented the use of PHCHRs with modifications to suit their population and culture. Two main types of PHCHR exist, namely the Child Personal Health Record (CPHR) and Maternal and Child Health Handbook (MCHH). While CPHR is confined to monitoring children health and/or development from birth, MCHH monitors mothers during pregnancy and the child from birth. This study focuses on the value of the two records, in terms of parents’ frequency of use/views of the PHCHR, as well as its impact on parent health knowledge and health-related outcomes.

Common major themes across the PHCHRs include keeping up-to date with immunisation, documenting diseases and health events, monitoring growth changes and developmental milestones, all of which play significant roles in child’s later life [1,2,3,6,7,8,9]. The PHCHRs are given to all parents during pregnancy [4] or at the birth of the baby [10], which enable periodic monitoring of children health and development. The central aim is to ensure optimum continuity of care [3], while at the same time serve as a means of communication between parents and professionals [11] and a public health education tool [2,6]. The records can be in an electronic form that is accessible to parents [12], or a hard-copy document that is kept by parents [3,10].

Various studies have investigated the individual PHCHRs with a wide range of scope, methods and outcome measures [1,3,4,6,13,14,15]. While these studies were successful in exploring the commitment of parents and/or professionals toward the use of the PHCHR or its effectiveness across different settings, limited evidence systematically summarises these findings. To our knowledge, only two studies systematically review parent-held child-related health records [16,17]. The earlier review [16] was aimed at meeting the information needs of patients (mothers) and healthcare providers in developing countries and concluded that such records offer insight into basic information needs of maternal-child healthcare providers. The second review [17] was confined to MCHH, which was associated with better mother’s knowledge, than practice in pregnancy and child care. The present review aims to examine whether the use of the PHCHRs (CPHR and MCHH) are valuable toward promoting child health and development.

## 2. Method

### 2.1. Review Protocol

The review followed the Preferred Reporting Items for Systematic Reviews and Meta Analyses (PRISMA) guideline [18]. A review protocol was developed and registered on PROSPERO (registration number: CRD42018096209).

### 2.2. Search Strategy

A systematic literature search was conducted from 1 April to 30 June in four primary databases; PUBMED/Medline, PsycINFO, CINAHL and Cochrane central register of controlled trials. This was extended with a Google Scholar search (first 10 pages) to identify any relevant electronic article that might not appear in the four primary databases. MeSH and index terms as appropriate, where developed to identify potentially relevant articles from MEDLINE, PsycINFO and Cochrane. These include the combination of (Parent-held OR parent acces* OR parent OR mother OR father OR caregiver) AND (record OR handbook OR child record OR child book OR record book) AND (monitor* OR improve* OR impact* OR affect* OR effect* OR determin*) AND (child outcome OR child health OR infant outcome OR infant health OR child develop* OR infant develop* OR development outcome OR child growth OR infant growth). For CINAHL and Google Scholar, the keywords ‘health records and child health, development’ were used in searching. An ancestral search of a reference list of articles that met the inclusion criteria from the databases was further undertaken to identify missing articles.

### 2.3. Eligibility Criteria and Selection

The inclusion criteria were studies of primary data that are in English language with full text available, assessed at least one child-related outcome (health/development), with parents/primary carers of children as participants, record commenced not later than the neonatal period, intended to be available through a government or NGO health strategy or education system. Study designs included are; Randomised Control Trials, quasi-experimental studies, single-arm studies and cross-sectional surveys that met the stated criteria. Studies were excluded if record books were not meant to record individual child health/development data. Two reviewers Muhammad Chutiyami and Shirley Wyver independently selected studies for inclusion. There was disagreement about one article meeting the inclusion criteria. The disagreement was resolved through discussion. List of excluded studies after full text screening, with reason for exclusion, is presented (Appendix A).

### 2.4. Data Extraction and Quality Assessment

Extraction and quality assessment were both conducted using a form adopted from Cochrane collaboration (Appendix A). The form consists of a total of 81 items out of which 71 to 80 were meant for assessing the risk of bias. For the purpose of this review, included studies were ranked by dividing the possible number of scores (0–10) by three ranking categories (low, medium, high), giving an approximate of four scores each. Hence studies were categorised as ‘Low Quality’ (0–3 items) ‘Medium Quality’ (4–7 items) and ‘High Quality’ (8–10 items) based on the number of criteria met, ranging from none (0) to all (10). Studies with medium to high quality were included in the analysis to ensure we only used studies with a low risk of bias. Extraction of all articles was conducted by the first author (Muhammad Chutiyami). Each article was assigned a number and we used a random number generator to select 20% of the articles for independent data extraction by the second author (Shirley Wyver). Extraction from Muhammad Chutiyami and Shirley Wyver was then checked for agreement.

### 2.5. Data Analysis

Meta-analysis of all findings was deemed inappropriate due to heterogeneity across outcomes reported by the included studies. Therefore, the study findings were first narratively synthesised, from which outcomes with sufficient data were pooled for a meta-analysis. Narrative synthesis (quantitative and qualitative evidence) was conducted in line with the study aim, based on the recommendation of the Centre for Reviews and Dissemination [19]. This involved a detailed examination of the numeric and textual summary findings, as well as conclusions reached in each study, with respect to the effectiveness of the record book (intervention) and parent use/views. The effectiveness was classified as ‘positive impact’ ‘no impact’ or ‘mixed impact’, taking into account the reported statistical significance and where necessary, the setting (Low and Middle-income countries and high-income countries) and quality of the study. Because there were few qualitative findings, quantitative findings were first synthesised under each respective outcome, followed by a summary of the qualitative findings as applicable. The broad outcomes were categorised as the impact of intervention (record book) on health-related outcomes, impact of intervention on parent health knowledge, parent use/views of the record book and frequency of data input in the record book.

Effectiveness outcomes with at least three studies that provided sufficient data for meta-analysis were pooled for meta-analysis. The analysis was conducted using Comprehensive Meta-Analysis software (CMA) version 3. Since considerable heterogeneity was expected, the meta-analysis was performed using a random-effects model with Odds Ratio (OR) as the pooled effect size measure. A *p*-value of < 0.05 is considered statistically significant. Studies that used a different measurement, including proportions, percentages, *t*-test and mean differences were accordingly converted to OR using the CMA. Q-statistic was calculated to indicate heterogeneous distribution of ORs between studies, while *I*^2^ was calculated to describe the percentages of total variation caused by heterogeneity across studies. Outcomes meta-analysed include the impact of PHCHR on child/maternal healthcare uptake, child immunisation uptake, utilisation of antenatal care, knowledge and practice of breast-feeding, and knowledge of pregnancy danger signs.

## 3. Results

### 3.1. Search Result

Of the four primary databases searched (MEDLINE, PsycINFO, CINAHL, Cochrane), a total of 6423 articles were identified in the first round. The articles titles/abstract were screened against the selection criteria and duplicates removed, after which 48 potentially relevant articles were identified. Google scholar search resulted in the identification of 22 additional articles that met the criteria based on title/abstract screening, making a total of 70 eligible studies. Of the 70 eligible articles, four articles full-text could not be retrieved while the 66 articles full-text were retrieved and screened against the eligibility criteria, of which 28 articles met all the criteria. These articles’ reference list was further searched, from which four more articles were identified making a total of 32 articles that fully met the criteria (Figure 1).

### 3.2. Characteristics of Included Studies

Based on World Bank [20] income level classification, nineteen of the studies were conducted in high income countries (Australia, France, Germany, Norway, New Zealand, Singapore, UK, U.S.), four in high middle-income countries (Brazil, South Africa), eight in low middle-income countries (Vietnam, Bangladesh, Palestine, Mongolia, Kenya, Indonesia, Cambodia) and one in a low income country (Uganda). For the purpose of analysis, low- and middle-income countries were categorised together as LMIC, while high-income countries were abbreviated as HIC. Except for 10 studies [21,22,23,24,25,26,27,28,29,30] that were conducted as a pilo*t*-test/trial to evaluate the efficacy/use of the child-related record, other studies were conducted as evaluation of existing records. All the studies were quantitative in nature, except one exclusively qualitative survey using an oral-history approach [31] and a few studies, which included a qualitative interview/focus group discussion section that supports the quantitative results [21,23,27,30,32,33]. The included studies were critically appraised for methodological quality/risk of bias using section of the tool adopted from Cochrane collaboration (Table 1). The mean quality score is 7.2, suggesting a medium quality for all included studies, with a range from 4/10 to 10/10. No study falls within the low-quality range (0–3), 17 studies were within the middle-quality range (4–7) while 15 studies were of high-quality range (8–10), out of which two RCTs conducted in Norway [22] and Indonesia [34] met all the 10 Criteria. Therefore, all the 32 studies were used in the analysis. Specific details of included studies were summarised in Appendix A.

### 3.3. Participants/Population

A total of 15,399 parents served as participants in the studies, ranging from 35 [31] to 1983 [44]. The majority of studies involved only mothers or primary carers (in the absence of mother) as the study participants [21,23,25,26,29,30,32,33,34,35,37,40,43,44,46,48,50,52], most of which were ≥20 years [21,23,30,33,34,35,36,44,48,50]. Males/Fathers were specifically mentioned as the respondents in only a few studies [31,36,45,47,49], although in small quantity (<20%). In one study, fathers were recruited but did not form part of the analysis [39]. Other studies mentioned parents/families as the respondents without giving a ratio of males to females. Most of the parents included, had at least a secondary/high school education [23,32,33,35,41,45,47,50] while other studies did not report or were unclear about educational level of the parents. The majority were categorised as having low [21,23,32] and middle-high [35,39,41,45] household/family/individual incomes. Other studies did not include or specify the income levels. Of the PHCHR analysed and/or children age considered, more than one third were at infancy/within first year of life [24,25,26,27,30,35,36,38,39,46,50,51], then smaller numbers for up to 2 years [28,32,34,44,47,52], three years [41,45,48] four years [33,42] and five years of age [22,29,40,43,49]. Age of children was not stated in one study [31] and unclear in some studies, which stated that record book was issued at early pregnancy stage/first ANC (antenatal care), then final outcomes assessed after 1 year [21,23] and after three years [37]. All studies were based on infants/children generally, except Emond et al. [38], whose study was based on a cohort of preterm versus term infants, and Koh et al. [45] study, which was based on children with developmental concerns versus no concern.

Health professionals were included as respondents in some studies, most of whom professionals involved in child health [23,25,27,36,42,46], GPs/Health visitors [39] or Nurses/midwives [30,31,40]. Yanagisawa et al. [30] included non-professionals, namely Village Health Volunteers (VHVs), as well as Traditional Birth Attendants (TBAs), who are involved in maternal and child care.

### 3.4. Intervention (PHCHR) Description

As noted previously, two types of PHCHR were used across the studies. These are the Child Personal Health Record (CPHR) and the Maternal and Child Health Handbook (MCHH). Of the interventions assessed, (24) were exclusively on Child Personal Health Record (CPHR) in high-, middle- and low-income countries, which focus on monitoring children health and/or development from birth [22,24,25,26,27,28,29,31,33,35,36,38,39,40,42,43,45,46,47,48,49,50,51,52]. The others were focused on Maternal and Child Health Handbook (MCHH) in middle-income countries, aimed at monitoring mothers during pregnancy and the child from delivery [21,23,30,32,34,37,41,44]. The impact/use of PHCHR was assessed using ‘intervention and control/comparison’ groups [22,26,28,30,37,44,46,52] or ‘pre- and pos*t*-test’ among similar populations [21,23,32]. Other studies assessed the PHCHR as a cross-sectional evaluation of parents/professional use of the records. Component of both approaches were interwoven and only identifiable in few studies [21,25,28,33,39,40,46,47,48], which focused mainly on parent support (e.g., important contact numbers, parent/child baseline, parent’s advice), parent education (e.g., immunization information, growth charts, developmental notes), child support/therapy (e.g., child vaccinations, health screening) for CPHR, and in addition, maternal/mother’s therapy (e.g., antenatal care, mothers vaccination, pregnancy drugs/screening) for MCHH.

### 3.5. Findings

#### 3.5.1. PHCHR Impact on Health-Related Outcomes

Twelve studies (n = 12) assessed the impact of PHCHR on nine health-related outcomes (Table 2), out of which four were pooled for meta-analysis (Figure 2). Out of six studies [22,23,26,30,34,44] that assessed PHCHR effect on the utilisation of child healthcare/MCH care, only two [30,44] showed an overall positive effect, and were statistically significant. Five of the six studies were pooled for meta-analysis (Figure 2a), which was not significant at 95% CI (OR = 1.31, 95% CI 0.92–1.88, *p* = 0.135, n = 5). Seven studies [21,23,26,34,44,46,47] assessed PHCHR effect on immunisation uptake, of which three showed positive effects [21,34,47]. Six of the seven studies provided sufficient data for meta-analysis about the impact of PHCHR on immunisation uptake, of which only one was from a HIC [46] and another one was on mother’s immunisation [34], hence were excluded from the analysis in order to estimate the pooled effect on child immunisation uptake in LMIC (Figure 2b). The pooled estimate shows a significant effect (OR = 1.93, 95% CI 1.01–3.70, *p* = 0.046, n = 4), suggesting parents who hold a PHCHR were approximately twice as likely to adhere to child immunisation uptake. Importantly, the largest effect size comes from a study with a high quality score (Table 1) from a low-income country [47]. Four studies [21,30,32,34] assessed the effect of the PHCHR (MCHH only) on the utilisation of antenatal care, all of which shows significant effects. The pooled effect (Figure 2c) indicates that parents who use the MCHH were approximately 1.6 times more likely to utilise recommended antenatal care (OR = 1.60, 95% CI 1.23–2.08, *p* = 0.000, n = 4). Three studies [21,32,34] assessed the effect of the PHCHR (MCHH only) on practice of breast-feeding/complementary feeding, all of which shows positive effect on breast feeding and one [34] indicated significant effect on complementary feeding for 6–9 month olds (OR = 4.35, *p* < 0.001). Pooled effect of the studies (Figure 2d) on breast feeding practice indicated a statistically significant effect (OR = 2.8, 95% CI 1.02–7.82, *p* = 0.046, n = 3), suggesting women who use the MCHH were about three times more likely to ensure adequate breast-feeding practice, including exclusive breast-feeding.

Only two studies [34,37] assessed the effect of PHCHR (MCHH) on child growth and development, both of which were of high quality (Table 1) and found statistically significant effect. Osaki et al., [34] reported significant effect of the MCHH on the risk of underweight (OR = 0.33, *p* < 0.05) and stunted growth (OR = 0.53, *p* < 0.05), both suggesting parents who hold the MCHH were less likely to have children with stunted growth/underweight. Dagvadorj et al., [37] on the other hand reported the impact of the MCHH on the risk of cognitive development (OR = 0.32, *p* = 0.007), suggesting a protective effect of MCHH on children risk of cognitive delay. Use of MCHH was also found to improve uptake of vitamin A among children [21,34], husband involvement in pregnancy/child care [23,34] and the use of family planning services [21] in LMIC. PHCHR effect on parent-professional communication showed no significant overall effect [22,23], although Hampshire et al., [39] reported better communication with health visitors compared to GPs (*p* = 0.008). Based on the current evidence, PHCHRs are only likely to be effective on health-related outcomes in LMIC, however, more studies are needed to establish a stronger cause-effect relationship.

#### 3.5.2. PHCHR impact on Parents/Mother’s Health Knowledge

Seven studies [21,22,23,30,32,34,44] reported intervention effects on parent’s health knowledge under nine outcome categorise (Table 3). None of the CPHR studies showed an effect on knowledge, all studies that suggested positive effect on knowledge used MCHH, and hence were from LMIC. Of the nine outcomes categorise, sufficient data for meta-analysis were available in two (Figure 3); knowledge of breast feeding and knowledge of pregnancy danger signs. Four studies [21,23,30,32] assessed the effect of intervention (MCHH) on knowledge of breast feeding, of which three shows overall positive effect but only one [32] is statistically significant. Pooled effect of the studies [21,23,32] indicated an odds ratio of 2.0, suggesting women who hold the MCHH are two times more likely to be knowledgeable about breast feeding. However, the effect is only slightly significant at 95% CI (1.004–3.985). Effect of the MCHH on awareness of pregnancy danger signs shows an overall pooled effect (OR = 3.16), which is not significant at 95% CI (0.93–10.78). Nevertheless, all the studies that assessed effect on knowledge of pregnancy danger signs [21,23,30,32] reported positive effect on awareness about early rupture of the membrane using different measurements (Table 3). Only one study shows positive (non-statistical) effect of MCHH on both knowledge of immunisation and family planning [21]. Similarly, one study each, shows statistically significant effect of intervention (MCHH) on knowledge of child sickness [34], ante-natal care [21] and general health knowledge (vaccination, danger signs, pregnancy risk factors and HIV/malaria prevention) [44]. While there are generally insufficient studies to conclude the effect of PHCHR on mother/parent knowledge, MCHH is likely to influence awareness on early rupture of the membrane and breast feeding in LMIC.

#### 3.5.3. Parent Views and Use of PHCHR

Over 70% (23/32) of the studies assessed different parent views and use of the PHCHR (Appendix A), which is well retained among the majority of parents, from six months [24,36] to five years [42]. More than half of parents read all/most content of the book [21,26,34,42,45] or read it to find useful information like phone numbers [29,36]. Reading habit significantly increased with educational level [23,41] and wealth index or prior explanation received [41] in LMIC. Proportion of parents (%) reporting similar outcomes (Appendix A) is expressed as ‘an average’ to give an approximate weight of the number of parents from different studies (n) reporting the same outcome. While the majority of parents (average 84%, range 15%–100%, n = 10) take the record book during regular visits to child clinics, fewer parents (average 52%, range 12%–81%, n = 7) take it to other child consultations like the GPs/specialists visits, all of which were reported in HIC that uses CPHR [22,27,28,33,36,39,42,46,52] except one study [40] from a LMIC. Significant correlation exists (*r* = 0.239, *p* = 0.009) between taking the record book to child clinics and GP visits [33], signifying parents who take the record to child clinics were more likely to take to it for GP consultations. Reported frequency of record input varied widely across the studies, which ranges from 21.6% [27] to 80.2% [39] of parents who always/usually make entries in the record book. Of the two studies that were conducted in LMIC using MCHH [21,41], high record input by parents (average 68%, range 59.8%–76.1%, n = 2) were reported, which is significantly influenced by high educational level (AOR = 2.16) and prior explanation received (AOR = 2.57) [41]. On the other hand, fewer record input by parents (average 43%, range 7.5%–80.2%, n = 8) were reported in HIC that uses CPHR [25,27,33,39,45,46,51,52].

Parents (average 72%, range 5%–100%, n = 14) generally like the PHCHR or consider it important for child care [21,22,24,25,26,27,28,36,39,40,42,45,46,51], which is more significant among teenage mothers and first-time mothers [39], and decrease with increasing child age [42]. This is an indication that young mothers are more likely to use/value the record, than mothers with previous pregnancy experience. A qualitative account of some parents (34 women and 1 man) considers the child record as a foundation towards establishing a long-term relationship between them and nurses, as well as a means of positive attributes between the mother and the child [31]. Parents give more preference to growth [28,33,39,40] and immunisation sections [40,42] among the different sections of the book, hence serve as a motive toward using the record book. Satisfaction with the PHCHR was high, with over 60% of parents, each giving a high rating to the PHCHR [22,52], while Stacy et al. [29] indicated moderate-high average rating without specifying the number of respondents. Similarly, qualitative interview with 20 mothers indicated their satisfaction with the book due to its appearance, practical information, convenience, long-term value, size and illustrations [30]. However, some parents (average 48%, range 25.1%–59.3%, n = 3) in both LMIC [21] and HIC [45,52] countries favours adding more illustrations/pictures to aid understanding.

##### Parent View of Professionals toward Use of the PHCHR

Parent reported less commitment of various health professionals toward the use of the PHCHR (Appendix A). Significant proportion of parents (average 40%, range 24%–56%, n = 2) indicated not being asked to present it [29,49], while 89% think the PHCHR will be more useful if professionals had shown interest [22]. Health visitors (Specialist community public health nurses in UK), were the most likely professionals reported to be using the PHCHR. Parents (average 93%, range 86.9%–100%, n = 4) reported a significant number of health visitors who regularly use the record book and/or put consultation information during visits [27,39,51,52]. This is closely followed by other nurses/midwives, whereby over two-third were reported by parents (average 70%, range 47%–92.5%, n = 4) as likely to use the record, particularly during regular child check-ups [33,42,51,52]. Hospital doctors were reported by parents (average 31.2%, range 5.8%–87.5%, n = 6) as the least likely to use the PHCHR, of which GPs use it [33,51] more than paediatricians [51] and casualty doctors [33,51]. Using the record is reported to be significantly higher among health visitors than GPs [39]. Similarly, public clinic staff were more likely to use the record than private clinic staff [40]. Therefore, community nurses are generally the most likely professionals to use/refer to the PHCHR in the healthcare setting.

#### 3.5.4. Record Input in PHCHR Measured by Direct Observation

Record input assessed through direct observation were reported in 18 studies (Appendix A), all of which were based on CPHR except two [21,32] that were on MCHH. Major sections reported include baseline, growth, development and immunisation data. Baseline information (maternal/parent data, child identification, birth weight, and APGAR scores) were all/mostly recorded across the studies [26,28,36,40,42,46] except two studies [48,50] which reported less than 50% completion rate. Growth data (weight and height) input range from 8.9% [48] to 90% [28]. Least recorded growth data were in LMIC, with only 8.9% growth and height charts [48] and 14% weight chart in a primary care setting [49]. Incompleteness of growth record was significantly associated with increased child age [35] and primary healthcare setting, compared to secondary/tertiary settings [49]. Development (milestones and checklists) record completeness varied from as low as 4.6% [35] to as high as 72% [24]. Incompleteness of development record is significantly associated with increase in number of children [35] while completeness of checklists is significant with English as first language [46]. Immunisation/vaccinations are well recorded (>60%) in most of the studies [24,25,28,40,42,43,48,49] which indicated less than 50% rate of completeness. The later could be related to the fact that the studies were conducted at the earliest stages of introducing the record, with less awareness about the use of it. Accurateness of immunisation record was checked by one study [43] that compared the record to national immunisation dataset, which showed 93% agreement (kappa = 0.42, *p* < 0.001). Proportion of general record input (without specification) was assessed in three studies [21,25,38], which range from 73.2% to 76.1%. Socio-economic variables were associated with general record input by parents, of which poor families (social index <25th centile) and Afro-Caribbean ethnic groups were significantly associated with fewer recorded information [38].

## 4. Discussion

This review examined the use of parent-held child health record (PHCHR) and its effect on child health and development. Thirty-one quantitative and one qualitative study met the inclusion criteria, all of which were of medium to high quality based on an assessment with a risk of bias tool adopted from Cochrane collaboration. The overall population size was 15,399 parents of children 0–5 years. Two parent-held child health record (PHCHR) were used across all included studies, which include the Child Personal Health Record (CPHR) and the Maternal and Child Health Handbook (MCHH). Both records were aimed at monitoring children health/development and in addition, the MCHH has a component that monitors women health during pregnancy. Similarly, all studies that assessed the MCHH were conducted in LMIC, and forms one-quarter of the 32 included studies. The findings of the review were reported under four broad outcome categories, which include the effect of the PHCHR on health-related outcomes (child and/or mother during pregnancy), the effect of PHCHR on parent health knowledge, parent views/use of the PHCHR and rate of data input in the PHCHR. The overall findings indicated that parent-held child health record is used and perceived by the majority of parents as important document for child care. However, there was no significant effect of the PHCHR on the utilisation of child/MCH care nor communication with professionals. Beneficial effect of the record on knowledge and health outcomes were in LMIC, whereby positive relationship was found with child immunisation uptake, vitamin A supplement uptake, child growth/development, knowledge and practice of both ante-natal care (ANC) and breast feeding, and awareness of pregnancy danger signs.

The finding of the review indicated a positive relationship between the use of the PHCHR and child immunisation uptake in LMIC. This may not be unconnected with the fact that resistance to immunisation in many HIC is quite low compared to LMIC [53,54,55]. Accordingly, both studies that assessed the effect of the PHCHR on immunisation uptake in the HIC [26,46] showed no overall effect. The positive association in LMIC could be as a result of poor awareness about immunisation, which characterised parents in many developing countries [4,56], of which the PHCHR is likely to educate parents about importance of immunisation. Although findings from this review indicated only one study [21] that reported the effect of the PHCHR on parent knowledge of immunisation, the study showed a positive effect. The finding of this study further shows beneficial effect of the PHCHR on child vitamin A supplement intake, a measure shown to be effective in reducing child mortalities [57,58]. Similarly, this study found that the use of PHCHR provided some protective effect against the risk of cognitive delay [37] and under-weight/stunted growth [34]. Both pieces of evidence were based on health promotion services uptake, notably ante-natal care visits and breast-feeding practice among mothers, which consequently affect child growth and development.

Ante-natal care (ANC) is another important preventive health service, particularly in developing countries, where maternal and child mortalities are high [58]. The finding of this study indicated a small, but significant relationship between the PHCHR (MCHH) and knowledge of recommended ANC [21], which support the review of Baequni et al. [17] that indicated a significant effect of the record book on knowledge of ANC. Similarly, this review showed that women who hold a PHCHR (MCHH) were about two times more likely to utilise the required ANC services. Poor utilisation of ANC is linked with adverse pregnancy outcomes, including preterm delivery, low birth weight and perinatal mortalities [59,60]. Therefore, providing women with PHCHR is likely to influence better pregnancy outcomes through the utilisation of recommended ANC. Another finding of this review indicated that PHCHR influence both knowledge and practice of breast-feeding among parents. This is not surprising as women who are more knowledgeable about the importance of breast feeding to the child, are more likely to put the knowledge into practice. Adequate breast feeding, particularly, exclusive breast-feeding within the first six months is associated with positive child health outcomes [61], hence the record book becomes valuable. This review further identified beneficial effect of the record book on awareness of pregnancy danger signs, particularly premature rupture of the membranes. Premature rupture of the membrane is a significant risk factor for placental infection and a cause of infant morbidities [62], which necessitates the need for early recognition and quick action. Therefore, the record book (MCHH) serves as a health education tool for women during pregnancy and the post-natal periods, which impact on infant outcomes.

This review also identified that the majority of parents from both HIC and LMIC use the record and take it to regular clinics check-ups. Similarly, proportion of data input in the record books were high, particularly baseline information and vaccinations, which are recorded by professionals, and growth/development data which are mostly recorded by parents. However, health professionals, particularly hospital doctors (GPs, paediatricians, casualty doctors) were reported as less likely to ask for the record or make inputs during consultations [22,29,49], which discourages parents toward the use of the record. Health personnel play an important role toward successful use of the health record book. Therefore, there is need for proper orientation among the professionals to be more committed in using the record, particularly, nurses and GPs who get in contact with the patients during regular visits. This in turn will results in the proper use of the record by parents, as they tend to value health information that comes directly from their healthcare providers [16].

This review has the strength of including all study designs, and using relatively larger number of studies (32) compared to the previous review [17] that is based on (5) included studies, there are some weaknesses/limitations that need to be taken into consideration. It should be noted that only 13 of the 32 studies were from LMIC, where the majority of the beneficial effects were identified. All meta-analysis results should be interpreted with caution because of the marked heterogeneity across the studies (Figure 1 and Figure 2), majorly due to differences in the included studies’ designs and method of assessing outcomes. Additionally, it should be noted that the effectiveness of the record on immunisation is not specific to a particular vaccine, but rather all reported child vaccinations. Therefore, effectiveness might differ between different childhood vaccinations. Furthermore, the meta-analysis was possible using only 8 of the 31 included quantitative studies, seven of which were from LMIC. This is so because most of the studies from the LMIC identified cause-effect relationship in comparison to the majority of studies from HIC, which assessed parent use/views. This might explain why the identified beneficial effect of the PHCHR was in LMIC.

## 5. Conclusions

Based on the findings of this review, it was concluded that parent-held child health record is valued and used by the majority of parents as an important document for child care, despite reportedly less commitment of health professionals. Its effects on health-related outcomes and parent knowledge is only likely in LMIC. The review was however unable to establish a strong meta-analysis result about parent health knowledge, but the record book showed beneficial effect on knowledge of breast feeding, ante-natal care and premature rupture of the membrane as a pregnancy complication. Similarly, a significant positive relationship was found towards child immunisation uptake, better breast-feeding practice and utilisation of recommended ante-natal care services in LMIC, all of which impact on child health and development. The findings from this review have implications not only in LMIC, where there is a marked positive effect, but in HIC, where the record is also valued and used by the majority of parents. The study thus recommends; more commitment of health professionals toward the use of the record, wide adoption with more studies to assess parent use/views of the record book in LMIC and finally, more studies that investigate cause-effect relationship in HIC.

## Figures and Tables

**Figure 1 ijerph-16-00220-f001:**
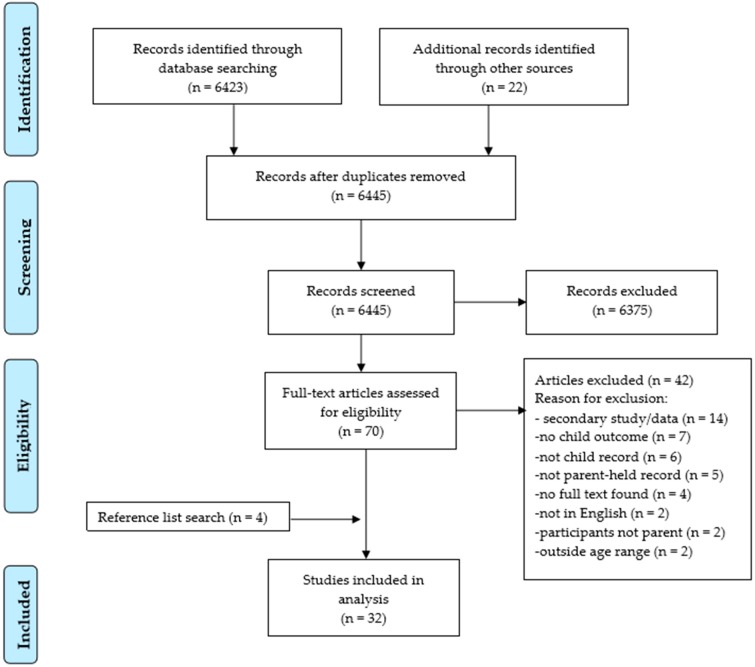
PRISMA (Preferred Reporting Items for Systematic Reviews and Meta Analyses) Flow chart for study selection.

**Figure 2 ijerph-16-00220-f002:**
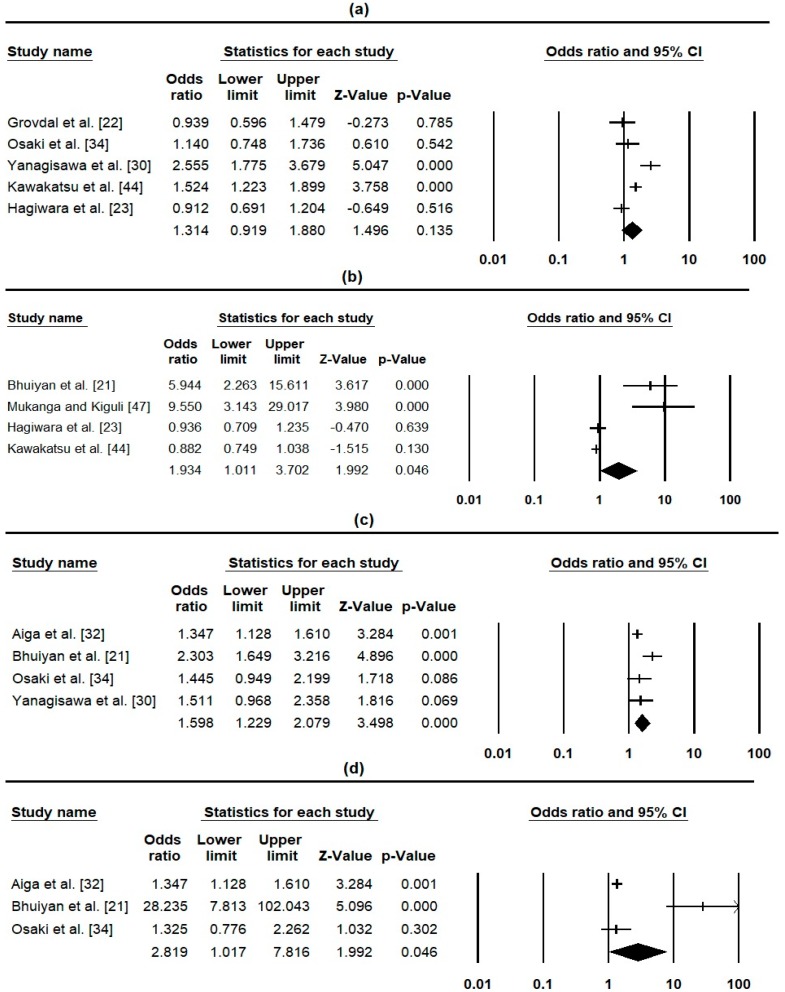
Forest plot of PHCHR effect on health-related outcomes: (**a**) Utilisation of child health care (Heterogeneity: Q-value = 23.65, *p* = 0.00, *I*^2^ = 83.09), (**b**) Uptake of child immunisation (Heterogeneity: Q-value = 31.19, *p* = 0.00, *I*^2^ = 90.38), (**c**) Utilisation of ante-natal care (Heterogeneity: Q-value = 7.75, *p* = 0.05, *I*^2^ = 61.31), (**d**) Practice of breast-feeding Heterogeneity (Heterogeneity: Q-value = 21.20, *p* = 0.00, *I*^2^ = 90.57).

**Figure 3 ijerph-16-00220-f003:**
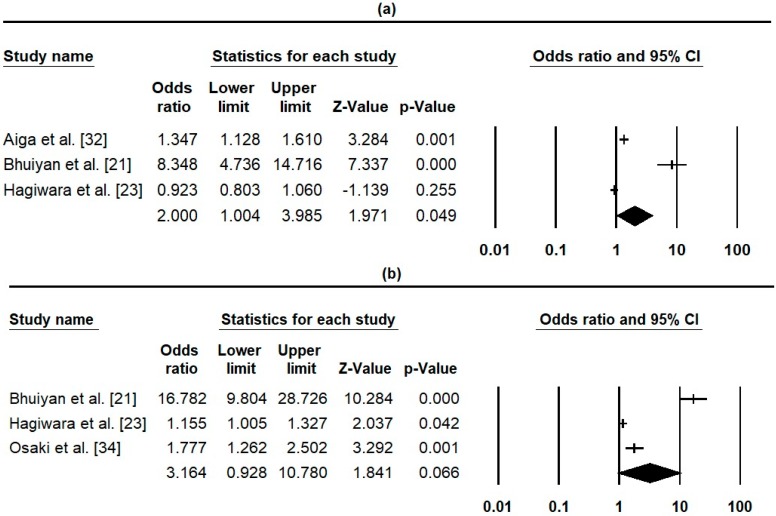
Forest plot of PHCHR effect on Parent Health Knowledge: (**a**) Breast Feeding (Heterogeneity: Q-value = 59.73, *p* = 0.00, *I*^2^ = 96.65), (**b**) Pregnancy danger signs (Heterogeneity: Q-value = 91.27, *p* = 0.00, *I*^2^ = 97.81).

**Table 1 ijerph-16-00220-t001:** Quality assessment/risk of bias.

Study	1	2	3	4	5	6	7	8	9	10	Total
Abud and Gaiva [35]	1	1	1	1	1	1	0	1	1	1	9/10
Aiga et al. [32]	1	1	1	1	1	1	0	1	1	1	9/10
Bhuiyan et al. [21]	1	0	1	1	1	0	0	1	1	1	7/10
Campbell and Halleran [36]	1	0	1	0	1	0	0	1	1	1	6/10
Clendon and Dignam [31]	1	0	0	1	1	0	0	1	1	1	6/10
Dagvadorj et al. [37]	1	0	1	0	1	1	1	1	1	1	8/10
Emond et al. [38]	1	0	1	1	1	0	0	1	0	1	6/10
Grovdal et al. [22]	1	1	1	1	1	1	1	1	1	1	10/10
Hagiwara et al. [23]	1	0	0	1	1	1	1	1	1	1	8/10
Hamilton and Wyver [33]	1	0	0	0	1	0	0	1	1	1	5/10
Hampshire et al. [39]	1	1	0	1	1	1	1	1	1	1	9/10
Harrison et al. [40]	0	0	0	1	1	0	0	1	1	1	5/10
Hikita et al. [41]	1	1	0	1	1	1	1	1	1	1	9/10
Jeffs et al. [42]	1	1	1	1	1	0	0	1	1	1	8/10
* Jessop et al. [43]	1	1	1	0	1	1	0	1	1	1	8/10
Kawakatsu et al. [44]	1	1	1	0	1	1	0	1	1	1	8/10
Koh et al. [45]	1	0	0	1	1	1	0	1	1	1	7/10
Lakhani et al. [46]	1	1	1	0	1	0	0	1	1	1	7/10
Miller [24]	1	0	0	0	1	0	0	1	1	0	4/10
Mukanga and Kiguli [47]	1	1	1	1	1	0	0	1	1	1	8/10
O’Flaherty et al. [25]	1	1	1	0	1	0	0	1	1	1	7/10
Osaki et al. [34]	1	1	1	1	1	1	1	1	1	1	10/10
Palombo et al. [48]	1	1	0	0	1	1	0	1	1	1	7/10
Polnay and Roberts [26]	1	0	0	0	1	1	0	1	1	1	6/10
Price et al. [27]	1	0	0	0	1	0	0	1	1	0	4/10
Saffin and Macfarlane [28]	1	1	0	1	1	0	0	1	1	1	7/10
Stacy et al. [29]	1	0	0	0	1	0	1	1	1	1	6/10
Tarwa and De Villiers [49]	1	0	0	1	1	0	0	1	1	1	6/10
* Troude et al. [50]	1	1	0	1	1	1	0	1	1	1	8/10
Walton and Bedford [51]	1	0	0	1	1	0	0	1	1	1	6/10
Wright and Reynolds [52]	1	1	1	1	1	0	0	1	1	1	8/10
Yanagisawa et al. [30]	1	1	1	1	1	0	0	1	1	1	8/10

Key: 1 = Yes—criteria met, 0 = No/Unclear/Not applicable. Areas assessed are numbered 1 to 10 on horizontal axis; 1—target population, 2—sampling frame, 3—sampling technique, 4—non-response bias, 5—data collection directly from subjects, 6—acceptable case definition, 7—instrument validity/reliability, 8—consistent data collection, 9—prevalence length of parameter of interest, 10—appropriate numerator/denominator of parameter of interest. * linkage studies—criteria 7 not applicable.

**Table 2 ijerph-16-00220-t002:** Impact of intervention (PHCHR) on health-related outcomes.

Outcomes	Reference	Intervention: CPHR, MCHH	Impact of Intervention on Outcome			Effect Size/Comments
			Positive	No Impact	Mixed	
Utilization of child healthcare/MCH care	[22]	CPHR		√		Non-routine CHC visits (66% intervention and 62% control *p* = 0.58), Doctor visits outside CHC (46% intervention and 48% control *p* = 0.95), Specialist/hospital visits (74% intervention and 73% control *p* = 0.84)
	[44]	MCHH	√			ATT on fever 0.095 (*t*-test = 4.024, *p* < 0.05) and diarrhoea 0.119 (*t*-test = 3.665, *p* < 0.001) (Treatment vs. control)
	[23]	MCHH			√	More time spent by women with first delivery at MCH consultations (*t*-test = 2.14, *p* ≤ 0.05) but no overall effect (*t*-test = 0.64), knowing next appointment date by primary educated women (*t*-test = 2.31, *p* ≤ 0.05) but not overall (*t*-test = 0.66)
	[34]	MCHH		√		Delivery with skilled birth attendant OR = 1.14
	[26]	CPHR		√		Hearing test visits: 76% experimental, 68% control. No child health clinic attendance: 1% experimental, 4% controls (both reported as not statistically significant)
	[30]	MCHH	√			Delivery with Skilled Birth Attendant (SBA), R = 12.2 (OR = 2.613, *p* < 0.01, AOR = 1.092), delivery in health facility, R = 4.4 (OR = 2.499, *p* < 0.01, AOR = 1.866, *p* < 0.01)
Uptake of immunization/vaccination	[21]	MCHH	√			8.3% case and 1.5% control (Child), 15.1% case and 6.6% control (mothers) (no *p*-value)
	[23]	MCHH			√	No overall effect on following child immunization schedules (*t*-test = 0.47), but effective among women with first delivery (*t*-test = 4.22, *p* ≤ 0.01)
	[44]	MCHH		√		Child vaccine ATT = 0.030, *t*-test = 1.516, (Treatment vs. control)
	[46]	CPHR		√		Diphtheria, Polio, Tetanus (DPT) 1st (90% study vs. 94% control), DTP 2nd dose (76% vs. 85%), DTP 3rd dose (22% vs. 31%), Pertussis 1st dose (64% vs. 58%), Pertussis 2nd dose (58% vs. 59%), Pertussis 3rd dose (19% vs. 19%)
	[47]	CPHR	√			Children with record compared to no record (OR = 9.55, 95% CI 3.19, 29.45, *p* < 0.001)
	[34]	MCHH	√			2 doses of tetanus toxoid (TT) (OR = 1.98, *p* < 0.01)
	[26]	CPHR		√		70% experimental and 65% control completed three vaccinations (reported as not statistically significant)
Utilization of antenatal care (ANC)	[32]	MCHH	√			67.5% pre-intervention 92.2% post intervention, *p* = 0.001 (≥3 ANC visits)
	[21]	MCHH	√			55.9% case and 35.5% control, *p* < 0.05
	[34]	MCHH			√	Significant with 6 ANC (OR = 1.67, *p* < 0.05) but not for 4 ANC (OR = 1.25) visits
	[30]	MCHH			√	R = 6.8 points—at least 1 ANC, R = 1.9—at least four ANC. Increased ANC attendance by four visits or more (AOR = 1.546, *p* < 0.05) but not significant with at least 1 ANC (OR = 1.476).
Practice of Breast Feeding (BF) and/or complementary feeding (CF)	[32]	MCHH	√			18.3% pre-intervention to 74.9% post-intervention, *p* < 0.001 (Exclusive BF)
	[21]	MCHH	√			16.6% case and 0.7% control for BF (no *p*-value)
	[34]	MCHH			√	CF for 6–9 months (OR = 4.35, *p* < 0.001), continuing BF (OR = 2.31, *p* < 0.01), continuing feeding with added proteins/vitamin (OR = 1.54, *p* < 0.05), continuing feeding with fruits/extracts (OR = 2.18, *p* < 0.001), continuing feeding with various snacks (OR = 4.14, *p* < 0.001), start BF, then continue CF in 6–9 months (OR = 2.7, *p* < 0.001), training self-feeding (OR = 2.75, *p* < 0.001). No significant effect (*p* > 0.05) for exclusive BF (OR = 0.76), feeding soft-rice thrice a day (OR = 1.29) and some kind of side dishes (OR = 1.35).
Child growth and development	[37]	MCHH	√			OR = 0.32 (protective effect) on the risk of cognitive delay (*p* = 0.007)
	[34]	MCHH			√	Less underweight children in case compared to control groups (OR = 0.33, *p* < 0.05), less stunted growth (OR = 0.53, *p* < 0.05) (adjusted for maternal BMI and birth weight of child). No significant difference in wasting growth (OR = 0.59, *p* > 0.05)
Uptake of Child vitamin A supplement	[21]	MCHH	√			17.6% case and 1.4% control (include both vitamin A and iron for mothers, no *p*-value)
	[34]	MCHH	√			Child vitamin A (OR = 2.0, *p* < 0.05)
Use of family planning	[21]	MCHH	√			41.5% case and 2% control (no *p*-value)
Parent-professional communication	[22]	CPHR		√		Communication with nurses (82% intervention and 77% control *p* = 0.66), doctors (75% intervention and 75% control *p* = 0.78), other professionals (83% intervention and 77% control *p* = 0.6)
	[23]	MCHH			√	Effective only for women with at least secondary education (*t*-test = 2.03, *p* ≤ 0.05) but no overall effect (*t*-test = 1.38) (child health discussion)
	[39]	CPHR	√			33.5% better communication with health visitors compared to GPs (24.6%), *p* = 0.008.
Mother-husband communication/husband support in MCH care	[23]	MCHH			√	Slightly effective for women with at least secondary education (*t*-test = 1.83, *p* = 0.1) but no overall effect (*t*-test = 1.15) (child health discussion)
	[34]	MCHH			√	Significant in husbands’ support with respect to saving money for delivery (OR = 1.82, *p* < 0.01), keeping their baby warm (OR = 1.58, *p* < 0.05), and giving their child developmental stimulation (OR = 1.62, *p* < 0.05). Not significant with Identifying blood donor (OR = 1.24), Acknowledging the expected date of delivery (OR = 0.93), Preparing transportation to delivery settings (OR = 1.03), Preparing home setting for delivery of child (OR = 0.75), Contacting health personnel (OR = 0.89), Supporting mother to breastfeeding, Bathing the infant/child (OR = 0.85), Caring cord of newborn (OR = 0.66), Bringing child to the healthcare facility (OR = 0.67), Bringing child to community-based integrated health post (OR = 0.7)

CPHR = Child Personal Health Record, MCHH = Maternal and Child Health Handbook, MCH = Maternal and Child Health, ATT = Average Treatment effect on Treated, CHC = Child Health Clinic, R = Point difference using ‘Difference-in-Difference’ analysis, OR = Odds Ratio, AOR = Adjusted Odds Ratio.

**Table 3 ijerph-16-00220-t003:** Impact of intervention (PHCHR) on parent health knowledge.

Outcomes	Reference	Intervention Type: CPHR, MCHH	Impact of Intervention on Outcome			Effect Size/Comments
			Positive	No Impact	Mixed	
Knowledge of child illness/MCH related conditions	[22]	CPHR		√		37% intervention and 47% control, *p* = 0.22 (child fever)
	[34]	MCHH			√	Mean (intervention/control) signs of: Newborn complications (1.64/1.84, *p* < 0.001, R = 0.29), sick child (1.93/2.28, *p* < 0.05, R = −0.28); preventing sick child (4.01 *p* < 0.001/3.68, *p* > 0.05, R = 0.56)
	[30]	MCHH	√			R = 6.2 points for anaemia, 9.9 for parasites, 7.5 for HIV transmission
Awareness on Breast feeding issues	[32]	MCHH	√			66.1% pre-intervention and 86.7% post-intervention, *p* < 0.001 (exclusive)
	[23]	MCHH			√	No overall effect (*t*-test = 1.14) but slightly significant for literate women (*t*-test = 1.85, *p* ≤ 0.1) (exclusive)
	[21]	MCHH	√			28.7% of case and 4.6% of controls (no *p*-value)
	[30]	MCHH	√			R = 6.2 for early breast feeding (no *p*-value)
Awareness of pregnancy danger signs/ Complications	[21]	MCHH	√			46.9% case and 5% control groups (no *p* value) (rupture of the membrane)
	[23]	MCHH	√			*t*-test = 2.04, *p* ≤ 0.05 (rupture of membra ne)
	[34]	MCHH	√			Mean (intervention/control): Pregnancy complications (1.63/1.46 *p* < 0.001, R = 0.49), delivery complications (1.42/1.56 *p* < 0.001, R = 0.41), postpartum complications (1.02/1.04, *p* < 0.001, R = 0.31), birth preparedness and complication readiness (1.30/1.24, *p* < 0.05, R = 0.54)
	[30]	MCHH			√	Effective; R = 12.4—swelling, 18.1—Persistent vomiting, 6—Severe headache/blurred vision, 2.8—Convulsion, 19.7—Bleeding from vagina, 11.3 Premature Rupture of the membrane, 5.9—Prolonged labour, 6.5—Malpresentation, 0.6—Placenta accrete, 3.8—Convulsions (no *p*-value), but not for Severe bleeding after birth (R = −5.1)
Knowledge of child development	[22]	CPHR		√		86% intervention and 79% control, *p* = 0.4
Knowledge of immunisation	[21]	MCHH	√			32.4% and 5.7% for case and controls (no *p*-value)
Knowledge of mother-child interaction	[22]	CPHR		√		52% intervention and 48% control, *p* = 0.84
Knowledge of family planning	[21]	MCHH	√			60.8% case and 5.0% control (no *p*-value)
	[23]	MCHH			√	No overall effect (*t*-test = 1.4) but effective among literates (*t*-test = 3.16, *p* = 0.01).
Knowledge of recommended Antenatal care	[32]	MCHH		√		91.9% pre-intervention and 93.7% post intervention, *p* = 0.1559 (visits ≥ 3)
	[21]	MCHH	√			78% case and 8.3% control groups, *p* < 0.05
General health knowledge	[23]	MCHH			√	No overall effect (*t*-test = 1.2) but effective with first delivery (*t*-test = 2.59, *p* = 0.01) (making oral rehydration salt)
	[44]	MCHH	√			ATT = 0.051 (*t*-test = 2.201, *p* < 0.05) (Treatment vs. control) (vaccination, danger signs, pregnancy risk factors and HIV/malaria prevention)

CPHR = Child Personal Health Record, MCHH = Maternal and Child Health Handbook, MCH = Maternal and Child Health, ATT = Average Treatment effect on Treated, R = Point difference using ‘Difference-in-Difference’ analysis.

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
