# Peer review of "Are Parent-Held Child Health Records a Valuable Health Intervention? A Systematic Review and Meta-Analysis"

_ijerph, 2019, doi:10.3390/ijerph16020220_

Round 1

Reviewer 1 Report

The manuscript was very well written. The authors did an excellent job describing the procedures, decision-making processes, and results of their meta-analysis. The presentation was professional and their results were appropriately interpreted. 

My only suggestion is a minor formatting issue. I think tables 4 and 5 are too long to be included in the body of the manuscript. I would prefer to have them included as appendices. 

Author Response

Thank you for your valuable suggestion. We have moved the tables to supplemental files.

Reviewer 2 Report

This is a complicated study due to complicated data outcome. Authors tried to summarize the effect of parent-held child health records on child health, but the results seemed more complicated due to complicated study design and outcome of studies involved in this review. However, this topic is very important and significant because the records recorded important information on maternal and child health and were also a public health intervention for promoting assess of women and children to health care services. Although many countries adopted this measure, a few assessments on maternal and child health care were found, especially on health of children. This study tried to evaluate this effect by systematic review. The following major problem should be addressed carefully. 1. in the part of methods, authors did not present what kind of studies, i.e. study deign, should be included. Why? When author conducted the meta analysis, they found high heterogeneity across the studies. A possible reason is that these studies used different study design. Pls think about this problem, it is suggested that some statement should be presented about this problem in this part. 2. Although there were lots of complicated outcome variables, the authors presented very little description about these outcomes. it is highly suggested that detailed statement about outcome studied should be given in the methods, especially, those used for meta-analysis, which will be helpful to make the manuscript more clear. 3. in the results, there were little information about intervention (PHCHR). Authors did not give detailed implementation of PHCHR for those studies involved in this study. Authors said that “ Impact/use of PHCHR was assessed using ‘intervention and control/comparison’ groups [22, 26, 28, 30, 37, 44, 46, 52] or ‘pre- and post-test’ among similar populations [21, 23, 32].” It means that authors assess the impact of PHCHR by using RCT studies, cross-sectional studies and pre-post studies in meta-analysis. I am not sure it is appropriate because they are different design. 4. Authors had a conclusion that “significant effect was found on child immunisation uptake, better breast-feeding practice and utilisation of recommended ante-natal care services in LMIC,…”. But in the part of methods, there were not any statement about classification about LMIC. If they decided to classify effect by country rank, it is better that some statement should be provided logically in advance. 5. it is suggested that some discussion or analysis statement about publication bias should be added in the part of discussion.

Author Response

Thank you for your valuable comments. 

1. in the part of methods, authors did not present what kind of studies, i.e. study deign, should be included. Why? When author conducted the meta analysis, they found high heterogeneity across the studies. A possible reason is that these studies used different study design. Pls think about this problem, it is suggested that some statement should be presented about this problem in this part.

Response: We have updated the eligibility criteria to include the study designs. We have acknowledged the high heterogeneity in the limitation paragraph (page 14)

2. Although there were lots of complicated outcome variables, the authors presented very little description about these outcomes. it is highly suggested that detailed statement about outcome studied should be given in the methods, especially, those used for meta-analysis, which will be helpful to make the manuscript more clear.

Response: we have updated the data analysis section by clearly identifying outcomes meta-analysed (Page 3). We mentioned the general outcomes in the data analysis and specified all the individual outcomes in the result tables in order to minimise the words count, as the manuscript stands around 6000 words without tables/figures.

3. in the results, there were little information about intervention (PHCHR). Authors did not give detailed implementation of PHCHR for those studies involved in this study. Authors said that “ Impact/use of PHCHR was assessed using ‘intervention and control/comparison’ groups [22, 26, 28, 30, 37, 44, 46, 52] or ‘pre- and post-test’ among similar populations [21, 23, 32].” It means that authors assess the impact of PHCHR by using RCT studies, cross-sectional studies and pre-post studies in meta-analysis. I am not sure it is appropriate because they are different design.

Response: Our research question and inclusion criteria were not restricted to any particular design, hence we analysed outcomes irrespective of the design, by sticking to the conclusions reached by the studies. We have also specified that outcomes assessed using measures other than an odds ratio, were accordingly converted to odds ration using the CMA (page 3). We would be happy to go with any other suggestion you may have in this regard.

We did not include the intervention implementation because it was not part of the outcomes this review covered (including the review protocol). Similarly, most of our included studies did not assess implementation as the programs have been in existence for long, but rather assessed use/impact.

4. Authors had a conclusion that “significant effect was found on child immunisation uptake, better breast-feeding practice and utilisation of recommended ante-natal care services in LMIC,…”. But in the part of methods, there were not any statement about classification about LMIC. If they decided to classify effect by country rank, it is better that some statement should be provided logically in advance.

Response: We have updated this section as per your suggestion (Page 3). We only give reference to the LMIC if all the studies that were used to reach that particular conclusion were from LMIC, otherwise outcomes were analysed irrespective of LMIC/HIC.

5. it is suggested that some discussion or analysis statement about publication bias should be added in the part of discussion. 

Response: We did not specifically assess publication bias, but rather a risk of bias of the included studies using a tool adopted from Cochrane collaboration. We have now made it clear in the discussion based on your recommendation (page 13) by indicating the tool we used to assess risk of bias.

Round 2

Reviewer 2 Report

the aruthors have addressed most of my comments. the manuscript has been improved. the following minor problems should be improved further. 1. in the methods, authors said "All study designs that met the stated criteria were included." It is still unclear about  design.  I suggest that the type of design should be presents rather than all design. such as RCT, Crossectional study, one -arrm study,etc 2. in the methods, authors said that "Studies that used a different measurement were accordingly converted to OR using the CMA". it is also unclear. what kind of measurements were converted to OR? Because not too many studies were included in meta-analysis, pls present them.

Author Response

Thank you for the valuable comments.

We have updated all the revisions accordingly.